# Survival and Functional Outcomes after Surgical Treatment for Spinal Metastasis in Patients with a Short Life Expectancy

**DOI:** 10.3390/jcm12010046

**Published:** 2022-12-21

**Authors:** Se-Jun Park, Chang-Hyun Ma, Chong-Suh Lee, Chung-Youb Jeon, Tae-Soo Shin, Jin-Sung Park

**Affiliations:** 1Department of Orthopedic Surgery, Spine Center, Samsung Medical Center, Sungkyunkwan University, School of Medicine, Seoul 06351, Republic of Korea; 2Department of Orthopedic Surgery, Haeundae Bumin Hospital, Busan 48094, Republic of Korea

**Keywords:** spinal metastasis, short life expectancy, survival rate, prognosis, functional outcome, surgical treatment

## Abstract

This study aimed to analyze the survival and functional outcome after surgery in spinal metastasis patients with a short life expectancy and to compare the baseline characteristics based on 3-month survival. A total of 492 surgical treatment cases with a preoperative revised Tokuhashi score **≤** 8were reviewed. Median survival was calculated and Kaplan–Meier analysis was used to analyze the survival rates at 6 months, 1 year, and 2 years postoperatively. The surgical period was divided into three time frames to examine the time trends. For the functional outcome, Eastern Cooperative Oncology Group Performance Status (ECOG-PS) was analyzed. This study categorized subjects based on 3-month survival and compared the baseline characteristics. The median overall survival was 10.6 months. The 2013–2020 period showed a significantly better median survival than the other two periods (*p* < 0.001). Lung and kidney cancers showed a significant survival improvement in 2013–2020 (*p* < 0.001). Patients with ECOG-PS **≤** 2 increased from 37.4% preoperatively to 63.7% postoperatively (*p* < 0.001). There were significantly more cases of preoperative favorable performance status, slow and moderate growth cancers, and chemotherapy after surgery in the survival ≥3 months group. Depending on the type of primary cancer, surgery can be considered even in spinal metastasis patients with a short life expectancy, particularly those with a good performance status.

## 1. Introduction

Spinal metastasis is the most common bone metastasis and occurs in more than 70% of terminal cancer patients [1,2]. Spinal metastases represent an important challenge for spine surgeons; metastasis surgeries are associated with significant morbidity and mortality [3]. In determining the treatment plan, decisions about the optimal treatment modality should be made carefully, and the patients’ life expectancy is considered one of the most important factors [4]. For many years, the essential prerequisite for any surgical indication for patients with spinal metastasis was an expected minimum life expectancy of at least 3 months [5,6,7,8]. For patients with a short life expectancy, surgical treatment is considered inappropriate because of the tendency towards lower functional improvement and the high risk of adverse events [9].

In recent years, the advancements in cancer pharmacotherapy, including chemotherapy, targeted therapy, adapted genotype treatment, and immunotherapy, have led to substantially improved patient survival in almost all cancer types [10,11]. Radiation and surgical technology improvements have also provided more treatment options to maintain or improve HRQOL, but there are few studies that reflect these advances [9]. We speculated that life expectancy predicted by the current prognostic scoring systems may be inaccurate for guiding the choice of surgical treatment, and the validity of these scoring systems is questionable for considering the effects of new therapeutic strategies on survival [8,12]. The actual life expectancy of the patient may be underestimated, preventing patients with a short life expectancy from benefiting from surgical treatment.

Therefore, the purpose of this study was to analyze the survival time and recent survival trends of spinal metastasis patients with a short life expectancy and to examine the functional outcomes. In addition, this study compared the baseline characteristics based on the actual 3-month survival.

## 2. Materials and Methods

### 2.1. Patient Selection

This study was performed retrospectively and approved by the Institutional Review Board in our institution. We enrolled patients who underwent surgical treatment for metastatic spinal tumor between 1997 and 2020 at a single institution. This study used a revised Tokuhashi score for predicting life expectancy of spinal metastasis patients, because Tabourel et al. reported that the Tokuhashi score showed the best sensitivity and specificity rates [11]. The criteria for patient selection were as follows: (1) patients whose preoperative revised Tokuhashi Score was 8 or less and for whom life expectancy was evaluated as less than 6 months [5]; and (2) patients whose functional outcomes were measured at 6 weeks postoperatively. Of the 798 patients who underwent surgical treatment for a metastatic spinal tumor in our institution, 306 patients and 62 patients were excluded because of the above two criteria, respectively. All data assessments were performed twice by 2 board-certified orthopedic surgeons (S.J.P. and J.S.P.) with a minimum of 10 years of experience each.

### 2.2. Surgical Indications and Methods

Surgeries were performed by spine surgeons at a single institution. The indications for surgical treatment were carefully determined through a multilateral, inter-department conference (medical oncologists, radiation oncologists, radiology specialists, and spine surgeons) based on the following criteria: (1) refractory pain despite conservative treatment; and (2) neurological deterioration or the potential for neurological deficits with spinal column instability. Four different surgical strategies were employed in the entire cohort: fixation only; palliative decompression and fixation; gross total removal and fixation. Patients who underwent percutaneous procedures, such as vertebroplasty or kyphoplasty, were excluded from the study. The surgical treatment method was determined based on the patient’s symptoms and radiologic imaging findings, which indicated the extent of tumor invasion. In cases of pathological fracture or mechanical instability without neurological deficits, only spinal stabilization was performed. Spinal stabilization was performed with a pedicle screw and rod-based system, using a standard open or minimally invasive method. Tumor removal was performed in cases of metastatic cord compression causing neurological deterioration. Palliative decompression was defined as the removal of only the tumor surrounding the cord for symptom relief. The maximum removal of tumors that invaded the vertebral body and surrounding cord was defined as gross total removal. The extent of tumor removal was determined based on the patient’s general condition and whether the primary malignancy was hypervascular or radioresistant. With the recent development of radiation therapy, the role of surgical treatment is aimed at relieving symptoms rather than resecting a large amount of tumor. Therefore, fixation is performed in the case of mechanical instability, and limited resection is performed only at the part of the tumor that causes cord compression and neurological symptoms. Remaining tumors are controlled with radiation therapy after the surgery. Therefore, the percentage of gross total removal has been decreasing in recent years (Figure 1).

### 2.3. Survival and Functional Outcome

This study analyzed the overall median survival time, survival rates, and follow-up survival rate at 6 months, 1 year, and 2 years postoperatively. The study cohort was divided into three groups for comparing the trend analysis according to the year of surgery, arbitrarily based on 8 years in chronological order from 24 years of surgical results: January 1997–December 2004; January 2005–December 2012; and January 2013–December 2020. This study also analyzed survival rates, 90-day mortality, follow-up at 6-months, 1-year, and 2-year survival rates and time trends of the top six common primary cancers (lung, liver, kidney, colorectal, breast and prostate cancer).

The patients’ performance status and functional outcomes were assessed according to the Eastern Cooperative Oncology Group (ECOG) performance status (0: asymptomatic; I: restricted physically; II: ambulatory and capable of all self-care; III: capable of only limited self-care; IV: completely disabled) [13] preoperatively and at 6 weeks postoperatively. The data were calculated and analyzed to determine whether there were any significant changes after surgery compared with before surgery. In addition, ECOG-PS was categorized to review the change of preoperative and postoperative status.

### 2.4. Analysis of Baseline Characteristics Based on 3-Month Survival

The baseline characteristics were examined based on actual 3-month survival. The patient factors were gender, age, and preoperative performance status (ECOG-PS). The primary cancer site was classified into three groups according to the growth rate (slow, moderate, rapid). The metastases to major internal organs, total number of vertebral metastases, and other bone metastases were evaluated to establish the burden of primary malignancy. The surgical factors included the type of surgical procedure (fixation only, palliative decompression and fixation, gross total removal and fixation). We also examined the use of both preoperative and postoperative adjuvant therapy (chemotherapy or radiotherapy). The total Tokuhashi score was analyzed to compare the expected prognosis between patients who survived more than 3 months and those who did not.

### 2.5. Statistical Analyses

Kaplan–Meier survival analysis was performed to examine the survival rates. A log rank test was performed to compare survival for the top 6 common cancers in the three time frames. Preoperative and postoperative overall functional outcomes were analyzed using a paired *t*-test. The chi-square test, Fisher’s exact test, and the independent *t*-test were used to compare the categorized values. Statistical analysis was conducted by professional statisticians using IBM SPSS software ver. 22.0 (IBM Corp., Armonk, NY, USA). *p* < 0.05 was considered statistically significant.

## 3. Results

### 3.1. Demographics and Baseline Data

A total of 492 patients with a predicted life expectancy within 6 months were included in this study, including 315 male patients (64.0%) and 177 female patients (36.0%). The mean age at spine surgery was 58.1 years (range 17.0–80.0). The thoracic spine was the most affected metastasis site, accounting for 59.8% (294/492) of all cases. Approximately 76.8% (378/492) of patients had primary cancers among the top six common cancers; lung was the most common primary cancer site (31.1%, 153/492) and prostate was the least common site (3.7%, 18/492). Spinal metastasis occurred most frequently in sites in the following order: lung; liver; kidney; colorectal; breast; and prostate. Metastatic disease was labelled as synchronous when the spinal metastases were diagnosed within a three-month interval of the diagnosis of the primary malignancy and metachronous if otherwise. Metastasis detection timing was metachronous in 77.8% (383/492) and synchronous in 22.2% (109/492). In terms of surgical year, from the number of patients with spinal metastasis who underwent surgery, it was identified that there was a tendency to increase over time (Table 1). The study population was examined according to each item of the revised Tokuhashi scoring system (Table 2).

### 3.2. Survival Analysis

The median overall survival time was 10.6 months (95% confidence interval (CI), 9.0–12.2) (Table 3). The survival rate according to the follow-up period was 66.8% (95% CI, 64.7–68.9) at 6 months, 46.1% (95% CI, 43.8–48.4) at 1 year, and 17.3% (95% CI, 15.5–19.1) at 2 years. The 90-day mortality rate was 12.0%. In terms of the common primary cancers, the median overall survival time was the highest in the breast group (24.2 months; 95% CI, range, 10.1–38.3) and the lowest in the colorectal cancer group (5.3 months; 95% CI, range, 0.5–10.1). The 6-months, 1-year, and 2-year survival rates were the highest in the breast group (94.1%, 82.1%, 50.5%, respectively), followed by the kidney group (84.0%, 57.7%, 27.5%, respectively). Colorectal cancer showed the lowest survival rate (46.8%, 29.2%, 0%, respectively). Colorectal cancer also showed the highest 90-day mortality of 16.7%.

Analysis of the three time frames showed that the median overall survival time was 7.0 months (95% CI, range, 4.4–9.6) for 1997–2004, 8.5 months (95% CI, range, 7.4–9.6) for 2005–2012, and 13.8 months (95% CI, range, 12.7–14.9) for 2013–2020 (Table 4). The recent 2013–2020 period showed a statistically significant improvement in survival time compared with the other two periods (*p* < 0.001) (Figure 2). Trends of survival in the top six common primary cancer sites were also analyzed. Lung and kidney showed statistically significant improvements in survival rates in 2013–2020 compared with the other two periods (*p* < 0.001). In the other cancers, the median survivals were improved, but no statistically significant difference was found.

### 3.3. Functional Outcome Analysis

The patients with ECOG-PS ≤ 2 increased from 37.4% preoperatively to 63.7% postoperatively. Overall, postoperative ECOG-PS showed a statistically significant improvement compared with preoperative outcomes (*p* < 0.001) (Figure 3). In the preoperative ECOG-PS 0–2 group, of 161 patients who were at least ambulatory and capable of self-care, 126 patients showed ECOG-PS improvement by at least one grade, with a rescue ratio of 78.3% (Table 5). In the preoperative ECOG-PS 3–4 group, of 269 patients who were capable of only limited self-care or were disabled, 139 patients showed improvement by at least one grade, with a rescue ratio of 51.6%. The preoperative ECOG-PS 0–2 group showed a better, improved rescue ratio compared with the 3–4 group and there was a statistically significant difference in the rescue ratio between the two groups (*p* < 0.001). However, in the preoperative ECOG-PS 0–2 group, 14 patients (8.7%) showed deterioration by at least one grade, and in the 3–4 group, 25 patients (9.3%) showed deterioration. There was no statistically significant difference in deterioration between the two groups (*p* = 0.491). 

### 3.4. Baseline Characteristics Analysis Based on Actual 3-Month Survival

There were no significant differences in gender, age, major internal organ metastasis, number of vertebral metastases, number of other bone metastases, surgical techniques, and total Tokuhashi score in the survival ≥3 months group than in the survival <3 months group (Table 6). However, in terms of performance status, there were significantly more cases of preoperative ECOG-PS 0–2 patients in the survival ≥3 months group than in the survival <3 months group (*p* = 0.021). In terms of primary cancer site, there were significantly more cases of slow and moderate categories of cancers in the survival ≥3 months group than in the survival <3 months group (*p* = 0.035). Compared with the survival <3 months group, the survival ≥3 months group had a significantly higher incidence of postoperative adjuvant chemotherapy treatment (*p* < 0.001). There were no significant differences in other adjuvant therapies, such as chemotherapy before surgery, radiotherapy before surgery, and radiotherapy after surgery, between the two groups (*p* = 0.129, 0.440, 0.427, respectively).

## 4. Discussion

Spine metastasis is quite common in patients with primary cancer [1,2]. Although the survival prognosis significantly depends on the primary cancer, most patients with spinal metastases are predicted to have a short life expectancy. Because life expectancy is an important factor in determining surgical treatment, there is sometimes hesitancy about active surgical treatment due to short life expectancy [9]. In addition, there are few studies on surgical treatment for patients with a short life expectancy. This study aimed to focus on the actual survival time and functional outcome after surgical treatment in spine metastasis patients with a short life expectancy. These data may provide more information to surgeons and patients with short life expectancy who require surgical treatment for pain and a low QOL.

Many authors have questioned the effectiveness of the current prognostic tools [8,9,11,14]. Most prognostic tools were reported in the 1990s to early 2000s, and few have been shown to be consistently reliable or useful in predicting survival [8,11]. In a retrospective study by Lee et al. [15], the authors found that the observed survival was much longer than the survival predicted by the revised Tokuhashi score. Such underestimation of life expectancy scores was also observed in other studies [16,17]. With advances in medical and surgical oncology, modalities with curative effects will increase, and these advances will lead to an improvement in the survival rate [11,14]. With these advances, it is more difficult to predict actual survival time accurately. From this point of view, we speculated that the current prognostic tools may have limitations in not reflecting the advances in oncology and they may be outdated, which underestimates actual life expectancy, leading to the exclusion of patients that would otherwise benefit from surgical treatment. We expected that the more recent it was, the longer the survival time would have been. In this study, the median overall survival time was 7.0 months (95% CI, 4.4–9.6) for 1997–2004 and 13.8 months (95% CI, range, 12.7–14.9) for 2013–2020. The survival time for 2013–2020 was significantly improved compared with the other two periods (*p* < 0.001).

Furthermore, among the top six common primary cancers, particularly in lung and kidney, there was a statistically significant improvement in survival for 2013–2020 compared with other periods. A number of studies reported that advances in maintenance treatment, such as targeted therapy, chemotherapy, and immunotherapy, have led to significant improvements in the overall survival of lung cancer and kidney cancer patients [18,19,20,21,22,23]. Our results indicated that advances in medical oncology in lung and kidney cancer were superior to those in other cancers over the past decades or that current treatments of other malignancies do not lead to significant improvements in survival rates compared with previous treatments. Further research is needed; however, if these results are substantiated, a more active palliative treatment can be considered in these patients.

Surgical treatment in patients with a life expectancy of less than 3 months to 6 months is controversial and the guidelines support performing surgical treatment in spinal metastasis patients with life expectancy of 3 months or more [6,9,24]. We investigated preoperative ECOG-PS and ambulatory status in patients with preoperative life expectancy of less than 6 months and measured outcomes at 6 weeks postoperatively. We detected a statistically significant improvement in functional outcomes postoperatively compared with preoperative outcomes. In terms of the rescue ratio, the preoperative ECOG-PS 0–2 group showed a better, improved rescue ratio compared with the 3–4 group. Our results indicated that if the patients had superior preoperative performance status, the postoperative functional recovery may also be superior. We thought that these results were meaningful. However, 14 patients (8.7%) showed deterioration of at least one grade in ECOG-PS 0–2, and in ECOG-PS 3–4, 25 patients (9.3%) showed deterioration. Alamanda et al. [25] suggested that patients should be made aware of the chance of loss of their ambulatory status and the possibility of being either wheelchair- or bed-bound as their disease course progresses. Although there was a statistically significant improvement in the functional outcome after surgery in our study, surgeons should also consider the possibility of deterioration of performance status.

The primary cancer site was classified into three groups according to the growth rate (slow, moderate, rapid) determined by the Tomita scoring system [26]. There were significantly more cases of slow and moderate categories of cancers in the survival ≥3 months group than in the survival <3 months (*p* = 0.035). Many authors reported the cancer type as a risk factor influencing survival, and our results were consistent with these studies [4,7,9,11,15,26,27]. In functional outcomes, there were significantly more cases of preoperative ECOG-PS 0–2 and ambulatory patients in the survival ≥3 months group than in the survival <3 months group (*p* = 0.021, 0.002, respectively). Dea et al. [9] reported that improvement in pain and HRQOL at 6 weeks postoperatively was more significant in patients who survived more than 3 months. These data and our results suggest that in patients with a short life expectancy, their actual life expectancy could be more, and surgical treatment may be considered to improve HRQOL. This conclusion is contrary to previous conclusions that patients with a short life expectancy do not benefit from surgical intervention. In addition, the survival ≥3 months group had a significantly higher incidence of postoperative adjuvant chemotherapy treatment (*p* < 0.001). We thought that may be due to the patient selection. Postoperative adjuvant chemotherapy cannot be performed on patients who are not in a physical condition to receive treatment; since these treatments are performed on patients in good physical condition after surgery, there may be a difference in survival. Other baseline characteristics showed no statistically significant difference in this study.

Our study has several limitations. First, as this is a retrospective study from a single hospital, it may have inevitable bias. Second, there were no data on the prognosis of the nonsurgical patients. This study aimed to determine the postoperative survival based on the timing of the need for surgical treatment. Therefore, this study provides additional information about survival and functional outcome to assist surgeons in their decision-making process. Lastly, this study focused on ambulation and self-care as key indicators to evaluate functional outcomes using ECOG-PS but did not use other indicators that can evaluate HRQOL, such as patient satisfaction, using questionnaires. Further studies will also be needed. However, all data were collected prospectively. In addition, this study included a relatively large number of patients who underwent surgical treatment at a single institution for 24 years, which is a relatively long time.

## 5. Conclusions

After surgery for spinal metastasis, which is associated with a short life expectancy of less than 6 months, the median survival was 10.6 months, which indicates that the actual survival was better than expected. This survival rate improved in recent years. Lung and kidney cancers showed a significant improvement in survival. Postoperative functional outcome was significantly improved. Depending on the type of primary cancer, surgical treatment can be carefully considered even in spinal metastasis patients with a short life expectancy considering the performance condition.

## Figures and Tables

**Figure 1 jcm-12-00046-f001:**
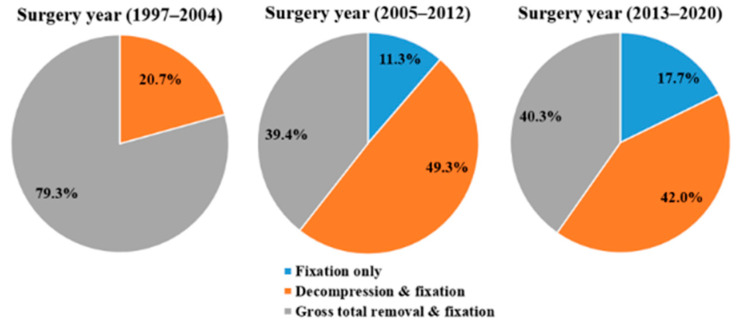
Trends in surgical methods for metastatic spinal tumors according to the year of surgery.

**Figure 2 jcm-12-00046-f002:**
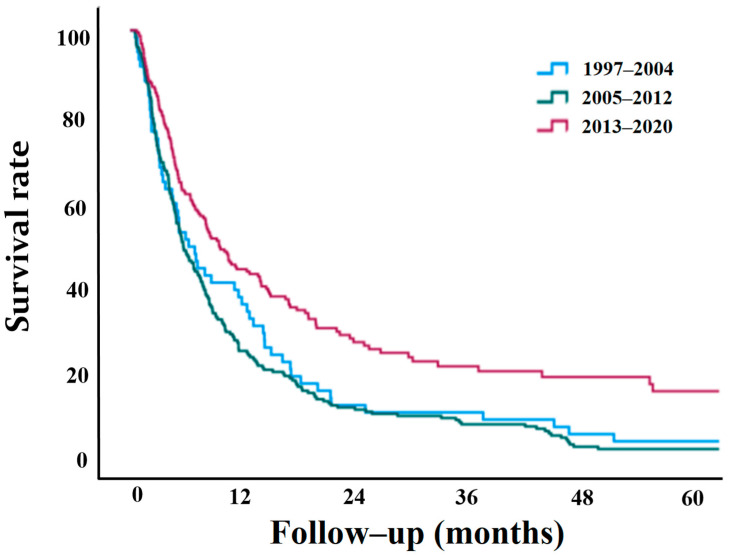
Differences in survival rates according to three-time frames (1997–2004, 2005–2012, 2013–2020).

**Figure 3 jcm-12-00046-f003:**
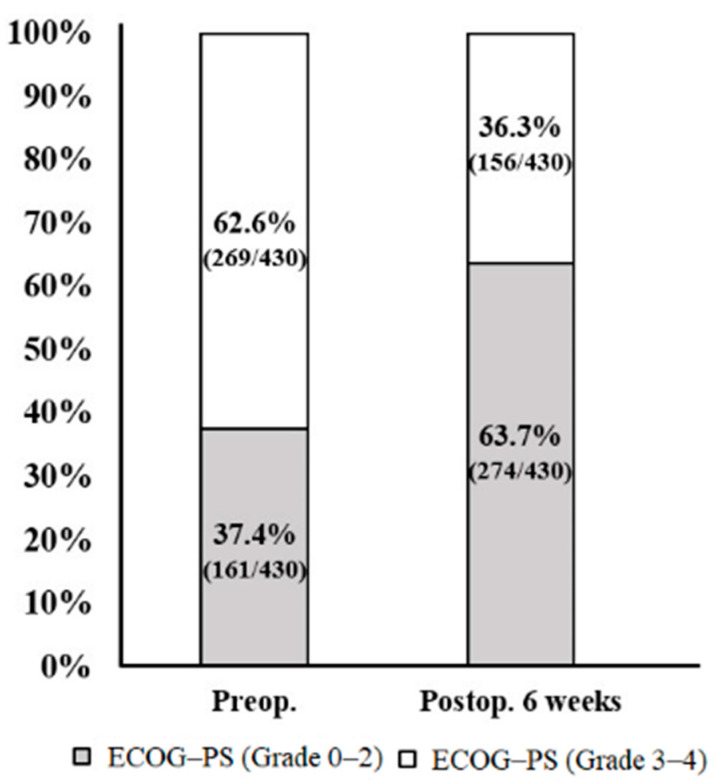
Preoperative and postoperative ECOG-PS and ambulatory status of patients.

**Table 1 jcm-12-00046-t001:** Baseline Demographics.

Characteristics, N	N = 492
Gender, N (%)	
Male	315 (64.0%)
Female	177 (36.0%)
Age at spine surgery, mean (range)	58.1 (17.0–80.0)
Male	58.9 (17.0–80.0)
Female	56.8 (26.0–79.0)
Location, N (%)	
Cervical	83 (16.9%)
Thoracic	294 (59.8%)
Lumbar	110 (22.3%)
Sacral	5 (1.0%)
Top 6 common primary cancer sites, N (%)	378 (76.8%)
Lung	153 (31.1%)
Liver	87 (17.7%)
Kidney	50 (10.2%)
Colorectal	36 (7.3%)
Breast	34 (6.9%)
Prostate	18 (3.7%)
Metastasis detection timing, N (%)	
Metachronous	383 (77.8%)
Synchronous	109 (22.2%)
Surgical technique, N (%)	
Fixation only	69 (14.0%)
Decompression and fixation	200 (40.7%)
Gross total removal and fixation	223 (45.3%)
Surgery year, N (%)	
1997–2004	58 (11.8%)
2005–2012	203 (41.3%)
2013–2020	231 (47.0%)

**Table 2 jcm-12-00046-t002:** Number of patients according to each item of revised Tokuhashi scoring system.

Characteristic		Score	N (%)
Performance status	Poor (10%–40%)	0	180 (36.6%)
	Moderate (50%–70%)	1	216 (43.9%)
	Good (80%–100%)	2	96 (19.5%)
No. extraspinal bone metastases foci	≥3	0	160 (32.5%)
	1–2	1	116 (23.6%)
	0	2	216 (43.9%)
No. metastases in the vertebral body	≥3	0	262 (53.3%)
	2	1	110 (22.4%)
	1	2	120 (23.4%)
Metastases to major internal organs	Unremovable	0	238 (48.4%)
	Removable	1	34 (6.9%)
	No metastases	2	220 (44.7%)
Primary cancer site	Lung, osteosarcoma, stomach, bladder, esophagus, pancreas	0	181 (36.8%)
	Liver, gallbladder, unidentified	1	100 (20.3%)
	Others	2	73 (14.8%)
	Kidney, uterus	3	52 (10.6%)
	Rectum	4	20 (4.1%)
	Thyroid, prostate, breast, carcinoid tumor	5	66 (13.4%)
Palsy	Complete (Frankel A, B)	0	49 (10.0%)
	Incomplete (Frankel C, D)	1	270 (54.9%)
	None (Frankel E)	2	173 (35.2%)

Total score: poor prognosis, 0–8; survival < 6 months, intermediate prognosis, 9–11; survival 6–12 months, good prognosis, 12–15; survival > 1 year.

**Table 3 jcm-12-00046-t003:** Overall survival data according to the top six common primary cancer sites.

Primary Cancer Site	N (%)	Median Months (95% CI)	6-Mo Survival Rate (95% CI)	1-Year Survival Rate, (95% CI)	2-Year Survival Rate, (95% CI)	90-Day Mortality, N (%)
All	492 (100%)	10.6 (9.0–12.2)	66.8% (64.7–68.9)	46.1% (43.8–48.4)	17.3% (15.5–19.1)	59 (12.0%)
Lung	153 (31.1%)	9.3 (7.2–11.4)	66.4% (62.6–70.2)	42.4% (38.3–46.5)	8.7% (6.3–11.1)	22 (14.3%)
Liver	87 (17.7%)	10.4 (7.0–13.8)	65.1% (60.0–70.2)	44.7% (39.3–50.1)	13.6% (9.2–18.0)	10 (11.5%)
Kidney	50 (10.2%)	14.2 (9.0–19.4)	84.0% (78.8–89.2)	57.7% (50.7–64.7)	27.5% (20.4–34.6)	2 (4.0%)
Colorectal	36 (7.3%)	5.3 (0.5–10.1)	46.8% (38.4–55.2)	29.2% (21.5–36.9)	0%	6 (16.7%)
Breast	34 (6.9%)	24.2 (10.1–38.3)	94.1% (90.1–98.1)	82.1% (75.5–88.7)	50.5% (41.7–59.3)	0 (0%)
Prostate	18 (3.7%)	8.5 (6.6–10.5)	66.7% (55.6–77.8)	42.4% (30.4–54.4)	24.2% (13.7–34.7)	2 (11.41%)

CI: confidence interval.

**Table 4 jcm-12-00046-t004:** Trends of survival in the top six common primary cancer sites according to the three time frames.

Primary Cancer Site	1997–2004 Median Months (95% CI)	2005–2012 Median Months (95% CI)	2013–2020 Median Months (95% CI)	*p*-Value
All	7.0 (4.4–9.6)	8.5 (7.4–9.6)	13.8 (12.7–14.9)	<0.001
Lung	5.0 (1.5–8.5)	8.6 (6.9–10.3)	13.1 (8.5–17.7)	<0.001
Liver	7.0 (0–17.7)	8.0 (5.7–10.3)	13.7 (10.3–17.1)	0.083
Kidney	6.0 (4.3–7.7)	7.6 (5.0–10.2)	25.6 (17.7–33.5)	<0.001
Colorectal	N/C	5.3 (4.3–6.3)	9.3 (2.8–15.8)	0.337
Breast	15.2 (5.6–25.0)	24.2 (0–50.2)	32.8 (0.4–65.2)	0.148
Prostate	N/C	8.5 (4.3–12.7)	12.6 (6.8–18.4)	0.640

N/C: not counted, CI: confidence interval.

**Table 5 jcm-12-00046-t005:** Preoperative and postoperative performance status based on ECOG-PS grade.

Preoperative ECOG-PS	Number of Patients (N)	Postoperative ECOG-PS				
		Grade IV	Grade III	Grade II	Grade I	Grade 0
Grade IV	102	44	16	31	11	0
Grade III	167	25	61	53	25	3
Grade II	121	4	3	13	71	30
Grade I	34	1	2	2	4	25
Grade 0	6	0	0	0	2	4
Total	430	74	82	99	113	62

Light gray box: aggravated performance status; white box: no change in performance status; dark gray box: improved performance status. ECOG-PS: Eastern Cooperative Oncology Group performance status; 0: asymptomatic; I: restricted physically; II: ambulatory and capable of all self-care; III: capable of only limited self-care; IV: completely disabled.

**Table 6 jcm-12-00046-t006:** Analysis of baseline characteristics based on actual 3-month survival.

Variable	Survival ≥ 3 Months Total: 433, N	Survival < 3 Months Total: 59, N	*p*-Value
Gender			0.976
Male	278 (64.2%)	38 (64.4%)	
Female	155 (35.8%)	21 (35.6%)	
Age			0.598
≥60	214 (49.4%)	27 (45.8%)	
<60	219 (50.6%)	32 (54.2%)	
Preoperative performance status(ECOG-PS), N (%)			0.021
0–2	178 (41.1%)	15 (25.4%)	
3–4	255 (58.9%)	44 (75.6%)	
Primary cancer site, N (%)			0.035
Slow: breast, thyroid, prostate, myeloma, lymphoma, endothelioma	76 (17.6%)	4 (6.8%)	
Moderate: kidney, colorectal, uterus, metastatic thymoma, tonsil, epipharynx	105 (24.2%)	11 (18.6%)	
Rapid: lung, liver, bladder, stomach, sarcoma, pancreas, malignant melanoma, unknown origin	252 (58.2%)	44 (74.6%)	
Major internal organ metastasis			0.156
Unremovable	211 (48.7%)	27 (45.8%)	
Removable	33 (7.6%)	1 (1.7%)	
No metastasis	189 (43.6%)	31 (52.5%)	
Number of vertebral metastases			0.180
≥3	224 (51.7%)	38 (64.4%)	
2	100 (23.1%)	10 (16.9%)	
1	109 (25.2%)	11 (18.6%)	
Number of other bone metastases			0.968
≥3	140 (32.3%)	20 (33.9%)	
1–2	103 (23.8%)	13 (22.0%)	
0	190 (43.9%)	26 (44.1%)	
Surgical technique, N (%)			0.747
Fixation only	61 (14.1%)	8 (13.6%)	
Decompression and fixation	174 (40.2%)	26 (44.1%)	
Gross total removal and fixation	198 (45.7%)	25 (42.4%)	
Adjuvant therapy, N (%)			
Chemotherapy before surgery	226 (52.2%)	37 (62.7%)	0.129
Chemotherapy after surgery	215 (49.7%)	15 (25.4%)	<0.001
Radiotherapy before surgery	214 (49.4%)	26 (44.1%)	0.440
Radiotherapy after surgery	222 (51.3%)	27 (45.8%)	0.427
Revised Tokuhashi score			0.093
Mean (95% CI)	5.3 (5.1–5.4)	4.8 (4.2–5.4)	

ECOG-PS: Eastern Cooperative Oncology Group performance status; CI: confidence interval.

## Data Availability

Not applicable.

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
