# Peer review of "Survival and Functional Outcomes after Surgical Treatment for Spinal Metastasis in Patients with a Short Life Expectancy"

_jcm, 2022, doi:10.3390/jcm12010046_

Round 1
Reviewer 1 Report
The Authors in this paper reported an article about the survival and functional outcomes after surgical treatment for spinal metastasis in patients with a short life expectancy.
The treatment of spinal metastasis and tools for the prediction of life expectancy are debated for several years. I agree with the authors about the advancements in cancer therapy and the actual life expectancy of the patient could be underestimated.
After careful review, I reported my consideration about the issue:
- The modified Tokuhashi score is the most used for this issue and several authors reported that the score is suggestive of actual survival for patients with a good prognosis. It is less accurate for patients with an estimated survival of less than 12 months. This could be a bias for this study and the authors should consider that.
- I am unable to visualize the table with more information about demographic populations (Tables are reported in the text but I have not seen them in the text). The section about methods is poor of date about details of the surgical procedure performed or the site of mts.
- The authors have not reported any demographic information about the population enrolled, distribution about the site of spinal metastasis, percentage or details about the type of surgery they did, and which segment of the spine was involved. I think that more information about this issue in the section methods could be a good improvement for the manuscript.
- They have devised the patients into three groups without any information about the details of the patients inside the group.
- I agree with the limitation reported by the authors. It is a retrospective study from a single institute, there was no data about the control group (nonsurgical).
- The authors could show some images about one or two cases to improve the quality of the manuscript.
Author Response
Thank you very much for your consideration of publication of our article and thank you for your useful comments. We would like to resubmit our revised manuscript and hope that it is now appropriate for publication in your journal. We tried to answer editor and reviewer's comments with utmost care. We revised our manuscript according to the helpful suggestions of the editor and reviewers and believe our manuscript is now much improved.
Reviewer’s comments:
Reviewer: 1
The Authors in this paper reported an article about the survival and functional outcomes after surgical treatment for spinal metastasis in patients with a short life expectancy.
The treatment of spinal metastasis and tools for the prediction of life expectancy are debated for several years. I agree with the authors about the advancements in cancer therapy and the actual life expectancy of the patient could be underestimated.
After careful review, I reported my consideration about the issue:
1. The modified Tokuhashi score is the most used for this issue and several authors reported that the score is suggestive of actual survival for patients with a good prognosis. It is less accurate for patients with an estimated survival of less than 12 months. This could be a bias for this study and the authors should consider that.
-Thank you for your comments.
As the reviewer comments about advancement of treatment technology for cancer, the use of prognostic scoring systems in the estimation of survival often leads to its underestimation and therefore carries the risk of certain patients not being treated.However, among the several currently available scoring system including revised Tokuhashi, Tomita, Modified Bauer, van der Linden, and Lei and Rades, the revised Tokuhasihi system was chosen because it showed the best sensitivity and specificity for predicting life expectancy [1]
1. Tabourel, G.; Terrier, L.M.; Dubory, A.; Cristini, J.; Nail, L.L.; Cook, A.R.; Buffenoir, K.; Pascal-Moussellard, H.; Carpentier, A.; Mathon, B.; et al. Are spine metastasis survival scoring systems outdated and do they underestimate life expectancy? Caution in surgical recommendation guidance. J Neurosurg Spine 2021, 35, 527-534. https://doi.org/10.3171/2020.12.SPINE201741.
2. I am unable to visualize the table with more information about demographic populations (Tables are reported in the text but I have not seen them in the text). The section about methods is poor of date about details of the surgical procedure performed or the site of mts.
- Thank you for your comments.
Result 3.1 is about demographics. The position of Table 1 has been modified.
The material method describes the indication and surgical method for spine metastasis. The general principles of spine metastasis surgery are to perform firm fixation if mechanical instability is presented regardless of the location of metastasis, and to remove the tumor if cord compression by the tumor is presented. The degree of tumor removal depends on the general performance of the patient and the vascularity and type of the tumor.
The case of removing only the tumor around the cord is called decompression. The case of removing as much tumor as possible up to the front of the vertebral body is called gross total removal. With the recent development of radiation therapy, the role of surgical treatment is aimed at relieving symptoms rather than resecting a large amount of tumor. Therefore, fixation is performed in the case of mechanical instability, and limited resection is performed only at the part of tumor that causes cord compression and neurological symptoms. Remaining tumors are controlled with radiation therapy after the surgery. Therefore, the percentage of gross total removal has been decreasing.
-> This information has been added to the material method (line 90-96). Information about surgical method was added in Table 1. Percentage (change) of surgical methods by year was added in Figure 1.
3 The authors have not reported any demographic information about the population enrolled, distribution about the site of spinal metastasis, percentage or details about the type of surgery they did, and which segment of the spine was involved. I think that more information about this issue in the section methods could be a good improvement for the manuscript.
- Thank you for your comments.
Information about the surgical method for spinal metastases was added in Table 1. Percentage (change) of surgical methods by year was added in Figure 1. General information on metastatic cancer surgery methods has been added to the material method (line, 90-96).
4. They have devised the patients into three groups without any information about the details of the patients inside the group.
- Thank you for your comments.
The study cohort was divided into three groups for comparing the trend analysis according to the year of surgery: arbitrarily based on 8 years in chronological order among 24 years of surgical results: January 1997–December 2004, January 2005–December 2012, and January 2013–December 2020.
-> We have describe the method to create three groups (line, 100-103)
5. I agree with the limitation reported by the authors. It is a retrospective study from a single institute, there was no data about the control group (nonsurgical).
- Thank you for your comments.
For 24 years at one research institute, this study reported survival outcome and functional recovery after surgery for metastatic spinal cancer in spinal metastatic cancer patients who were predicted to have a poor survival prognosis of less than 6 months. The purpose of this study is to focus on not withdrawing the surgical treatment according to survival expectancy by a scoring system that evaluates the prognosis of the current patient. This study shows that the assessment of the remaining life expectancy can be underestimated and surgery can be considered depending on the type of cancer and the patient's performance. Therefore, comparing the results with the control group (nonsurgical), which is not indicated for surgical treatment, is unlikely to be meaningful.
6. The authors could show some images about one or two cases to improve the quality of the manuscript.
- Thank you for your comments.
Although the reviewers’ opinion are fully respected, for spinal metastatic cancer, the prognosis and functional recovery after surgery depend on the factors such as the primary cancer type, the patient's performance status, and the effectiveness of chemotherapy rather than the surgical type or method. Since the images of an example case do not appropriately carry this information (the primary cancer type, the patient's performance status, and the effectiveness of chemotherapy), inserting such images may diminish or blur the emphasis of the paper.
Thank you very much for reviewing our manuscript.
We are looking forward to hearing a positive reply from you.
Sincerely

Reviewer 2 Report
Expert review "Survival and functional outcomes after surgical treatment for spinal metastasis in patients with a short life expectancy"
I thank you for the excellent opportunity to read this article.
General Statement and Summary
I read with great interest and attention the article "Survival and functional outcomes after surgical treatment for spinal metastasis in patients with a short life expectancy" submitted for publication in the journal JCM.
In this article, the authors conducted a retrospective analysis of patients (n=492) who underwent surgery between 1997 and 2020 at a single medical institution for tumor metastasis to the spine.
The purpose of this study was to analyze survival duration and recent survival trends of patients with spinal metastases with short life expectancy and to examine functional outcomes in metastasis of the most common six primary tumors (lung, liver, kidney, colon, breast, and prostate cancers)
The overall period was divided into three time intervals 1997-2004, 2005-2012, and 2013-2020. The division was to qualitatively determine the trend of postoperative survival duration. Patients' health status and functional outcomes were assessed using (ECOG) preoperatively and 6 weeks postoperatively.
The median overall survival was 10.6 months. In terms of common primary cancers, median overall life expectancy was highest in the breast group (24.2 months; 95% CI, range, 10.1-38.3) and lowest in the colorectal cancer group (5.3 months; 95% CI, range, 0.5-10.1). Analysis of the three time frames showed that the median survival duration was 7.0 months in 1997-2004, 8.5 months in 2005-2012, and 13.8 months in 2013-2020. The overall postoperative ECOG-PS showed a statistically significant improvement over preoperative outcomes. These data make it clear that the preoperative survival estimates for patients with short life expectancy after surgery for spinal metastasis may differ from their actual life expectancy.
The advantage of this article is the considerable amount of accumulated material in one hand.
The authors were also honest about their limitations. This is a retrospective study within a single hospital, which may have study bias. There was no data on the prognosis of non-surgical patients. And only ECOG-PS was used as a key indicator and assessment of functional outcomes, but no other indicators were used.
Specific comments.
In order to improve the perception of the publication, a number of changes should be made:
The authors need to add figures (survival time statistics by annual periods, results of functional analyses) and tables (demographics, trend of patients with metastases by years, total life expectancy, dynamics of survival by annual periods, ratio of groups with survival ≥ 3 months and < 3 months by total Tokuhashi score by clinical and demographic indicators) in the attachment, this will greatly improve the perception of this work.
Author Response
Thank you very much for your consideration of publication of our article and thank you for your useful comments. We would like to resubmit our revised manuscript and hope that it is now appropriate for publication in your journal. We tried to answer editor and reviewer's comments with utmost care. We revised our manuscript according to the helpful suggestions of the editor and reviewers and believe our manuscript is now much improved.
Reviewer 2
Expert review "Survival and functional outcomes after surgical treatment for spinal metastasis in patients with a short life expectancy"
I thank you for the excellent opportunity to read this article.
General Statement and Summary
I read with great interest and attention the article "Survival and functional outcomes after surgical treatment for spinal metastasis in patients with a short life expectancy" submitted for publication in the journal JCM.
In this article, the authors conducted a retrospective analysis of patients (n=492) who underwent surgery between 1997 and 2020 at a single medical institution for tumor metastasis to the spine.
The purpose of this study was to analyze survival duration and recent survival trends of patients with spinal metastases with short life expectancy and to examine functional outcomes in metastasis of the most common six primary tumors (lung, liver, kidney, colon, breast, and prostate cancers)
The overall period was divided into three time intervals 1997-2004, 2005-2012, and 2013-2020. The division was to qualitatively determine the trend of postoperative survival duration. Patients' health status and functional outcomes were assessed using (ECOG) preoperatively and 6 weeks postoperatively.
The median overall survival was 10.6 months. In terms of common primary cancers, median overall life expectancy was highest in the breast group (24.2 months; 95% CI, range, 10.1-38.3) and lowest in the colorectal cancer group (5.3 months; 95% CI, range, 0.5-10.1). Analysis of the three time frames showed that the median survival duration was 7.0 months in 1997-2004, 8.5 months in 2005-2012, and 13.8 months in 2013-2020. The overall postoperative ECOG-PS showed a statistically significant improvement over preoperative outcomes. These data make it clear that the preoperative survival estimates for patients with short life expectancy after surgery for spinal metastasis may differ from their actual life expectancy.
The advantage of this article is the considerable amount of accumulated material in one hand.
The authors were also honest about their limitations. This is a retrospective study within a single hospital, which may have study bias. There was no data on the prognosis of non-surgical patients. And only ECOG-PS was used as a key indicator and assessment of functional outcomes, but no other indicators were used.
Specific comments.
In order to improve the perception of the publication, a number of changes should be made: The authors need to add figures (survival time statistics by annual periods, results of functional analyses) and tables (demographics, trend of patients with metastases by years, total life expectancy, dynamics of survival by annual periods, ratio of groups with survival ≥ 3 months and < 3 months by total Tokuhashi score by clinical and demographic indicators) in the attachment, this will greatly improve the perception of this work.
- Thank you for your comments.
For 24 years at one research institute, this study reported survival outcome and functional recovery after surgery for spinal metastatic cancer patients who were predicted to have a poor survival prognosis of less than 6 months. This study showed that the actual survival rate of spinal metastasis patients who were assessed to have a poor survival prognosis of less than 6 months was actually better than expected. Therefore, the purpose of this study is to focus on not withdrawing the surgical treatment according to survival expectancy by a scoring system that evaluates the prognosis of the current patient. Additionally, overall period was divided into three time interval to show that the survival rate of patients with spinal metastasis has recently improved. Although the reviewers’ opinion are fully respected, it is not the purpose of this study to report the annual survival rates and functional recovery results for 24 years. Also, it would be difficult to show meaningful results in terms of time or number of subjects. This study analysed prognostic factors that differed between the two groups according to 3-month survival after surgery in order to evaluate the factors that affect actual survival outcome. Analysis of various prognostic factors including Tokuhashi socre between the two groups was performed.
-> To improve the perception of the paper, the authors added general information about the surgical method for spinal metastasis to the Material and Method Section (Line, 90-96). Also, changes in the surgical method for spinal metastasis by time period is added in Figure 1.
Thank you very much for reviewing our manuscript.
We are looking forward to hearing a positive reply from you.
Sincerely

Round 2
Reviewer 1 Report
Ok good article